# Content of Phenolic Compounds in Meadow Vegetation and Soil Depending on the Isolation Method

**DOI:** 10.3390/molecules25225462

**Published:** 2020-11-22

**Authors:** Anna Ziolkowska, Bozena Debska, Magdalena Banach-Szott

**Affiliations:** Department of Biogeochemistry and Soil Science, University of Science and Technology, 6 Bernardynska St., 85-029 Bydgoszcz, Poland; anna.ziolkowska@onet.eu (A.Z.); debska@utp.edu.pl (B.D.)

**Keywords:** meadow soils, acid hydrolysis, alkaline hydrolysis, phenolic compounds, HPLC

## Abstract

The aim of this paper was to determine the effect of the hydrolysis method on the amounts of phenolic compounds in the plant material in soil and, as a consequence, on the parameters to determine the degree of lignins transformation in soils. The study included the plant material (hay, sward, and roots) and soil—Albic Brunic Arenosol (horizon A, AE, and Bsv) samples. Phenolic compounds were isolated at two stages by applying acid hydrolysis followed by alkaline re-hydrolysis. The quantitative and qualitative analysis of phenolic compounds was performed with high-performance liquid chromatography with a DAD. The content of phenolic compounds in the extracts depended on the hydrolysis method and it was determined by the type of the research material. The amounts of phenolic compounds contained in the alkaline hydrolysates accounted for 55.7% (soil, horizon Bsv)—454% (roots) of their content in acid hydrolysates. In the extracts from acid hydrolysates, chlorogenic and *p*-hydroxybenzoic acids were dominant. In the alkaline extracts from the plant material, the highest content was recorded for *p*-coumaric and ferulic acids, and in the extracts from soil, ferulic and chlorogenic acids. A combination of acid and alkaline hydrolysis ensures the best extraction efficiency of insoluble-bound forms of polyphenols from plant and soil material.

## 1. Introduction

Phenolic acids and phenolic aldehydes represent a group of polyphenols, common in plant tissues. They are secondary metabolites with a varied chemical composition and biological properties [1,2]. In plants, they are mostly found in bound forms, as esters or glycosides, as lignin and tannin hydrolyzing components [1]. As for the chemical composition, phenolic acids, as lignin derivatives, present in plant cells can be divided into groups, which differ in the number and the place of substitution of hydroxyl and methoxyl groups:−Hydroxybenzoic acid and its derivatives, e.g., dihydroxybenzoic, protocatechuic (PA), salicylic (SA), syringic (SYR), and vanillic (VA) acids;−Hydroxycinnamic acid and its derivatives, e.g., cinnamic, ferulic (FEA), *p*-coumaric (*p*-CO), and caffeic (CA) acids.

Another group of phenolic compounds are depsides of the molecule core, which includes an ester bond, e.g., chlorogenic acid (CHA) is produced as a result of a combination of the carboxyl group of caffeic acid with the phenolic group of quinic acid [2].

Phenolic compounds can be isolated from solid samples using many different methods, e.g., simple extraction, extraction combined with hydrolysis, ultrasound, or microwave radiation-assisted extraction [1,3,4]. Very often, the simple extraction of phenolic compounds using organic solvents shows a low performance as it mostly allows for isolation of free phenolic compounds and the bound forms can be diregared. To isolate the insoluble-bound forms, it becomes justifiable to apply preliminary hydrolysis, followed by extraction [1,3].

Verma et al. [4] and Irakli et al. [5] reported on the use of the Folin-Ciocalteu reagent to facilitate assaying of the total values of both free and bound phenolic acids. Verma et al. [4] and Ross et al. [6] reported on the isolation of phenolic acids from plant material samples, with the selection of the type of the hydrolysis applied (acid, alkaline) being essential. Verma et al. [4] demonstrated that higher contents of *p*-coumaric, vanillic, and sinapic acids were recorded as a result of the alkaline hydrolysis, as compared with the contents of the compounds produced from acid hydrolysis, whereas, as for syringic acid (SYR), an inverse dependence was recorded. Besides, the authors show that some phenolic compounds are soluble in an acid environment and some in an alkaline one.

Phenolic compounds isolation and assaying in plant material were also covered by Ross et al. [6]. They compared three sample hydrolysis methods: alkaline hydrolysis followed by acid hydrolysis, alkaline hydrolysis alone, and acid hydrolysis alone. Each method comes in two variants: the first one involved adding ascorbic acid (AA) and ethylenediaminetetraacetic acid (EDTA), while in the second variant, no solutions of those acids were added. As a result, the highest performance was noted after the application of alkaline hydrolysis with AA and EDTA, as compounds protecting phenolic compounds against decomposition were added. The effect of adding AA and EDTA on the stability of phenolic compounds has also been confirmed by Nardini et al. [7]. The results reported by Verma et al. [4] and Ross et al. [6] demonstrate that, to receive the total contents of phenolic acids and aldehydes found in the research material sample, both types of hydrolysis should be applied and AA and EDTA should be added.

The papers that report on a comparison of the hydrolysis method for plant material and soils are missing. Assaying the contents of phenolic compounds in soils facilitates the evaluation of the degree of plant residue decomposition in soil [8,9,10,11,12,13]. Besides, determining at least the mutual proportions of phenolic compounds of the syringyl, vanillyl, and cinnamyl type makes it possible to determine the genus of the plant undergoing decomposition in soil [11,12,14,15].

The aim of this paper was to determine the effect of the hydrolysis method on the amounts of phenolic compounds recorded in the plant material (hay—H, sward—St, root—R) and soils (horizons A, AE, and Bsv) and, as a result, on the parameters, to find the degree of lignins transformation in soils.

## 2. Results

In the extracts from acid hydrolysates and alkaline re-hydrolysates from the plant material and soil samples, 11 phenolic aldehydes and phenolic acids were identified: protocatechuic acid (PA), *p*-hydroxybenzoic acid (*p*-HA), chlorogenic acid (CHA), vanillic acid (VA), syringic acid (SYR), caffeic acid (CA), vanillin (VAN), syringaldehyde (SYAL), *p*-coumaric acid (*p*-CO), ferulic acid (FEA), and salicylic acid (SA) (Appendix A).

There was considerable variation in the content of respective phenolic compounds, which occur in the extracts from acid hydrolysates and alkaline re-hydrolysates and the plant material and soil samples. Sample chromatograms for the separation of the phenolic compounds contained in the extracts after acid hydrolysis and alkaline re-hydrolysis are presented in Figure 1.

### 2.1. Contents of Phenolic Compounds in the Plant Material Samples

The quantitative analysis of the phenolics of the hydrolysates from the plant material has shown that in the extracts from alkaline re-hydrolysates, significantly higher contents of the total phenolic compounds analyzed were recorded, as compared with their contents in the extracts from acid hydrolysates (Table 1). The average contents of the total aldehydes and phenolic acids in the extracts from alkaline re-hydrolysates were 4900.3 (hay), 3598.5 (sward), and 2585.2 µg/g d.w. (roots), which accounted for 75.9%, 81.0%, and 84.7% of the phenolic compounds isolated as a result of both types of hydrolysis. The extracts from acid hydrolysates showed higher contents of *p*-hydroxybenzoic acid (*p*-HA) than the extracts from alkaline re-hydrolysates. Besides, from the hay and sward samples, while applying the method of acid hydrolysis, more chlorogenic acid (CHA) was produced than after alkaline re-hydrolysis. CHA in the extracts after acid hydrolysis accounted for 84.4% and 68.9% of its total from both hydrolyses. In the extracts from acid hydrolysates from the root samples, 70.7 µg/g d.w. less CHA was assayed than in the extracts from alkaline re-hydrolysates.

The mean contents of the other phenolic compounds, in the extracts from alkaline re-hydrolysates, were significantly higher, as compared with their mean contents in the extracts from acid hydrolysates. The biggest differences in terms of the mean content of the phenolic compounds analyzed were recorded for *p*-coumaric acid (*p*-CO). The differences were 1509.2, 1379.1, and 874.2 µg/g d.w. for H, St, and R, respectively. When applying the acid hydrolysis, only 8.1%, 3.2%, and 1.9% of the total content of *p*-CO was isolated. With the alkaline re-hydrolysis method, high contents of ferulic acid (FEA) were recorded, which for H, St, and R were, on average, 1988.0, 1092.2, and 732.5 µg/g d.w. and accounted for 90.8%, 94.9%, and 95.9% of the content of that acid in both hydrolysates. The differences in the content of vanillic aldehyde (VAN) in the H, St, and R samples across extracts were 185.0, 157.2, and 90.2 µg/g d.w. The method of acid hydrolysis allowed assaying of 22.6%, 17.9%, and 16.6% of the total contents of VAN, respectively. The contents of salicylic acid (SA) in the alkaline re-hydrolysates from hay, sward, and roots were 106.0, 129.0, and 73.2 µg/g d.w., which accounted for 64.3%, 91.0%, and 88.4% of the content of the acid assayed with the two methods. The extracts from alkaline re-hydrolysates contained 248.2, 134.9, and 96.7 µg/g d.w. of syringic aldehyde (SYAL), which accounted for 80.3%, 69.2%, and 70.7% of the total contents of this aldehyde. The differences in the contents of caffeic acid (CA) across the extracts of hay, sward, and roots were 102.4, 49.5 and 82.4 µg/g d.w. The method of alkaline re-hydrolysis allowed for assaying 72.2%, 62.1%, and 68.2% of the total amounts of CA produced as a result of both hydrolyses. The share of vanillic acid (VA) in the extracts from alkaline re-hydrolysates from hay, sward, and roots accounted for 69.1%, 73.2%, and 87.9% of the total VA contents.

### 2.2. Contents of Phenolic Compounds in Soil Samples

The extracts from acid hydrolysates and alkaline re-hydrolysates of the soil sampled 5 and 25 m away from the irrigation ditch from three horizons (A, AE and Bsv) demonstrated a considerable variation in the mean contents of phenolic compounds (Table 2). The total contents of all the phenolic aldehydes and phenolic acids analyzed in the extracts from alkaline re-hydrolysates of the samples of soil from horizon A were, on average, 310.4 µg/g d.w. higher than in the extracts from acid hydrolysates. The alkaline re-hydrolysis method facilitated assaying 70.6% of the total amounts of phenolic compounds isolated in those samples applying both hydrolyses. Only the mean contents of vanillic (VA) and caffeic (CA) acids and both aldehydes (VAN, SYAL) in the extracts from acid hydrolysates and alkaline re-hydrolysates were similar. The contents of the other compounds, namely *p*-CO, SA, FEA, PA, CHA, *p*-HA, and SYR, in the extracts from alkaline re-hydrolysates accounted for 96.4%, 83.8%, 79.1%, 76.0%, 75.4%, 69.6%, and 65.7% of the total of both types of hydrolyses, respectively.

In the soil sampled from the eluvial horizon (AE) and illuvial (Bsv) horizon, the total content of phenolic compounds in the extracts from acid hydrolysates, as compared with the alkaline re-hydrolysates, was higher. The differences were 51.2 (AE) and 96.0 µg/g d.w. (Bsv), respectively. With the acid hydrolysis, 55.2% and 64.2% of the total phenolic compounds produced as a result of both hydrolyses, respectively, were recorded.

For the soil sampled from horizon AE, the highest difference across the extracts was recorded for *p-*hydroxybenzoic acid (*p*-HA). The share of *p*-HA extracted with the method of acid hydrolysis accounted for 87.6% of its total content recorded applying both methods. The extracts from acid hydrolysates also showed higher contents of protocatechuic (PA) and vanillic (VA) acids and syringic aldehyde (SYAL), as compared with the content in the extracts from alkaline re-hydrolysates. No significant differences were demonstrated in the content of chlorogenic acid (CHA), syringic acid (SYR), caffeic acid (CA), and vanillic aldehyde (VAN) between acid hydrolysis and alkaline re-hydrolysis. The contents of ferulic (FEA), *p*-coumaric (*p*-CO), and salicylic (SA) acids were higher in the solutions after alkaline re-hydrolysis than in the solutions after acid hydrolysis. The share of those compounds in alkaline re-hydrolysates accounted for 84.2%, 72.6%, and 69.5% of the total of those compounds produced as a result of both hydrolyses, respectively.

For the soil sampled from the illuvial horizon (Bsv) in the extracts of acid hydrolysates, *p*-hydroxybenzoic acid (*p*-HA) was identified most. Its content in acid hydrolysates accounted for 81.7% of the total of both hydrolyses. In the extracts, the lowest difference in the content was noted for protocatechuic acid (PA); its share in acid hydrolysates accounted for 61.6% of the total from both hydrolyses. The method of alkaline re-hydrolysis, on the other hand, facilitated the isolation of 75.0%, 68.8%, and 64.9% of the total amounts of salicylic (SA), *p*-coumaric (*p*-CO), and ferulic (FEA) acids.

## 3. Discussion

The contents of phenolic compounds recorded in the samples of plant and soil material demonstrated that the application of alkaline hydrolysis after acid hydrolysis considerably increased the amount of the phenolic compounds extracted. As stressed earlier by Mroz et al. [1], Verma et al. [4], Ross et al. [6], Nardini et al. [7], and Krygier et al. [16], the importance of the hydrolysis method selected on the quality and the amount of the phenolic compounds was reported.

The literature most frequently presents the results of research on the application of acid hydrolysis to isolate phenolic compounds from the samples of plant material and soils [1,6,13,17,18]. Verma et al. [4], however, note the importance of applying alkaline re-hydrolysis after acid hydrolysis in terms of the amount of the phenolic compounds produced. After alkaline re-hydrolysis, Verma et al. (2009) [4] and Kim et al. [19] recorded even a few-fold higher amounts of ferulic and *p*-coumaric acids, as compared with the amounts assayed in the samples after acid hydrolysis alone. Additionally, the other identified phenolic compounds point to a possibility of producing their higher amounts, which was confirmed in the research presented in this paper.

For all the samples of plant material (hay, sward, roots) and soil sampled from horizon A, higher total contents of phenolic compounds were recorded, as a result of alkaline re-hydrolysis, as compared with acid hydrolysis alone (Table 1 and Table 2). Interestingly, for the plant material and soil of horizon A, the alkaline re-hydrolysis method provided the grounds for producing ferulic (FEA) and p-coumaric (*p*-CO) acids and, as for sward, roots, and soil of horizon, as well as for salicylic acid (SA). The share of FEA, *p*-CO, and SA in alkaline re-hydrolysates of horizons AE and Bsv was also higher than in the acid hydrolysates. Verma et al. [4] reported on acid hydrolysis facilitating a release of considerably higher amounts of salicylic acid than alkaline re-hydrolysis. However, due to the differences in the research material, and mostly the chemical composition of the materials compared, one cannot expect that the dependencies will be identical.

In the acid hydrolysates of the root and the soil sampled from horizons AE and Bsv, *p*-hydroxybenzoic acid was dominant. The samples of hay and sward, on the other hand, provided considerably higher amounts of *p*-hydroxybenzoic and chlorogenic acids, as compared with their amounts isolated with the alkaline re-hydrolysis.

A significantly higher share of phenolic compounds in the extracts after acid hydrolysis, as compared with alkaline hydrolysis for the soil sampled from horizons AE and Bsv, can be a consequence, next to the tissue structure and a decreasing amount of organic matter of plant origin, deep down the soil profile, of a change in the proportions of the compounds of soil organic matter. The higher the degree of organic matter humification, the higher the share of hydrolyzing compounds in the acid solution, as compared with the compounds hydrolyzing in the alkaline solution. In horizon A, the presence of the extended meadow vegetation root system [20] affects the amount of organic matter undergoing decomposition, and thus the content of phenolic compounds, especially those isolated in the alkaline reaction solution.

To sum up, it must be stressed that, to avoid a risk of underestimating the content of phenolic compounds that occur in the material studied, if possible, two types of hydrolysis should be applied.

Isolating and identifying phenolic compounds allow for the division of those compounds into three groups referred to as:−Vanillyl compounds (V), as the total content of vanillin and vanillic acid derived from coniferyl alcohol;−Syringyl compounds (S), as the total content of syringaldehyde and syringic acid, derived from sinapyl alcohol;−Cinnamyl compounds (C), as the total content of ferulic acid, *p*-coumaric acid, and caffeic acid, derived from coumaryl alcohol (Figure 2 and Figure 3).

The ratios of those compounds in lignins are plant species specific. It was also shown that, despite a considerable transformation of lignins during plant material humification, determining the ratio of the compounds allows for determining their origin [8,9,12,13,14,15]. As reported by Kovaleva and Kovalev [12], lignins of herbaceous plants contain higher amounts of cinnamyl compounds than the contents of vanillyl and syringyl compounds. The lignins of grasses and meadow herbs can contain from 4- to 6-fold more cinnamyl compounds than the lignins of trees, and the V:S:C ratio for the aboveground part of grass vegetation of sub-alpine meadows was 1:1:6. As reported by, e.g., Crawford [21], angiosperm lignins contain approximately equal amounts of vanillyl and syringyl compounds and little cinnamyl compounds. Gymnosperms lignins show a definite advantage of vanillyl compounds over the other ones.

In this paper, the content of vanillyl, cinnamyl, and syringyl compounds and their share in the V+S+C pool were calculated to verify the effect of the method of isolating phenolic compounds on the parameters studied. The parameters calculated for the contents of phenolic compounds in acid hydrolysates and for the sum of both hydrolyses (acid hydrolysis + alkaline hydrolysis = sum (total) hydrolyses) were compared.

Irrespective of the methods of assaying the phenolic compounds (acid hydrolysis, sum of both hydrolyses) applied, the contents of vanillyl, syringyl, and cinnamyl compounds were definitely higher in the plant material, as compared with their contents in soil. Interestingly, regardless of the sample type, cinnamyl compounds were dominant, and no significant differences in the content of vanillyl and syringyl compounds were found (Figure 2 and Figure 3).

The share of cinnamyl compounds in the V+S+C pool (Figure 4), following the application of both hydrolyses in the plant material (hay, sward, roots), ranged from 78.87% to 81.65%, and in soil from 51.49% to 61.5%, whereas after the application of acid hydrolysis only, the share of cinnamyl compounds accounted for 49.93–64.24% and 41.12–47.23%, respectively (Figure 4). It is obviously a consequence of a significant effect of the method of alkaline hydrolysis on the efficiency of the extraction of *p*-coumaric and ferulic acids.

As seen from Table 3, irrespective of the method applied, the ratio of vanillyl to syringyl compounds was, in general, 1:1. The V:S:C parameter, following the application of both hydrolyses for plant material, ranged from 1:1:8 to 1:1:9 and for soil, from 1:1:2 to 1:1:3, whereas, after the application of acid hydrolysis only, it was 1:1:2–1:1:4 and 1:1:1.5–1:1:2, respectively. However, it must be stressed that, irrespective of the method of assaying the contents of phenolic compounds, one records the dependencies, which allows formulation of identical conclusions for the changes that occur in the process of plant material humification in soils.

## 4. Materials and Methods

### 4.1. Chemicals

HCl, NaOH, ascorbic acid, EDTA, ethyl acetate: (all of analytical grade), and acetonitrile, methanol, and acetic acid (all of HPLC grade) were purchased from Avantor Performance Materials (Poland).

The analytical standards of phenolic compounds were purchased either from Sigma-Aldrich (Steinheim, Germany): protocatechuic, *p*-hydroxybenzoic, chlorogenic, caffeic, ferulic and salicylic acids, and vanillin; or from Fluka (Steinheim, Germany): vanillic, syringic and *p*-coumaric acids, and syringaldehyde.

The water was purified using the Millipore Milli-Q system.

### 4.2. Materials

The samples were taken from Poland’s and Europe’s unique perennial grasslands complex, the Czerskie Meadows (53°87′ N; 18°12′ E). The Czerskie Meadows Complex was established in the mid-19th century and it is located in various parts of the Tuchola Forest (Poland). In total, it covers an area of about 2000 ha. The uniqueness of this complex is related to the irrigation system, namely the slope-and-flood system applied back then, which, in most areas of the Czerskie Meadows, has still survived.

The samples were taken in 2011 from the plot where irrigation had been stopped since 1996, referred to by the founders as the Green Meadow. The plant material (hay, sward, and roots) and soil (horizons A, AE, and Bsv) were sampled from three specified stands. At each stand, the soil was sampled 5 and 25 m away from the irrigation ditch. The soil was sampled from 6 soil profiles (a total of 18 plant material samples and 18 soil samples). The soil samples were dried in room temperature and sieved (2 mm). The plant material samples were dried in room temperature and ground. In total, 18 plant material samples and 18 soil samples were analyzed, whereas the results provided in the tables are mean values for the three stands specified and two distances (5 and 25 m). The soil in all the profiles was analyzed in compliance with the WRB systematics as *Brunic Albic Arenosol*. The basic soil properties are presented in Table 4 [21].

### 4.3. Methods

#### 4.3.1. Extraction of Phenolic Compounds from the Samples of Plant Material and Soils

To isolate the phenolic compounds from the plant material and soils, two types of hydrolysis were applied:−Acid hydrolysis with a 6 M HCl solution; and−Alkaline re-hydrolysis with a 10 M NaOH.

The details of the extraction procedures are given in Figure 5.

#### 4.3.2. Qualitative and Quantitative Analyses of Extracts Containing Phenolic Compounds

The organic phase of phenolic extracts was evaporated until dry and then dissolved in 2.5 mL of MeOH. The qualitative and quantitative composition of phenolic compounds contained in the extracts from hydrolysates from the samples of the plant material; the samples of soils were assayed using the high-performance liquid chromatographer HPLC Series 200 by Perkin–Elmer (Shelton, CT, USA) equipped with a DAD. The analytic column, Bionacom Velocity STR (Genore Chromatography, Warsaw, Poland), with 5 μm in particle diameter and 250 × 4.6 mm in size was used.

The mobile phase consisted of:−Eluent A: H_2_O:CH_3_CN:CH_3_COOH (88.5:10:1.5—% V); and−Eluent B: CH_3_CN.

The samples for analysis were prepared by combining 0.5 mL filtered through a nylon filter, φ = 0.45 µm of the extract of phenolic compounds with 0.5 mL of eluent A. The injection volume was 10 µL (three replications). The gradient division program was applied at a rate flow of 1.3 mL/min. The initial composition of the mobile phase was 100% of eluent A applied. The content of eluent B grew linearly from the 14th min of the analysis and it reached 10% in 36 min, then decreased, reaching the final stage of the analysis (in the 42nd min) at 0%. The detection was made at the wavelength of 280 nm. The pattern of the chromatogram for the reference solution of phenolic compounds and their retention times (Figure 6) facilitated identification of the following phenolic compounds: protocatechuic acid (PA), *p*-hydroxybenzoic acid (*p*-HA), chlorogenic acid (CHA), vanillic acid (VA), syringic acid (SYR), caffeic acid (CA), vanillin (VAN), syringaldehyde (SYAL), *p*-coumaric acid (*p*-CO), ferulic acid (FEA), and salicylic acid (SA) (Appendix A). The content of phenolic compounds is given in µg/g d.w. The quantitative analysis of the phenolic compounds identified was performed using the external reference curves of the dependence of the peak area for the concentration of the phenolic compound (μg/mL). The reference response curve for each phenolic compound was a linear regression with regression coefficients > 0.98. The detailed data are provided in Table 5. The HPLC analysis of reference solutions was performed as for the phenolic extracts.

### 4.4. Statistical Analysis

The results were statistically verified by determining the following: mean, maximum and minimum, standard deviation (SD), and standard error (SE). The significance of the differences between the content of vanillyl, syringyl, and cinnamyl compounds for each variant (hay, sward, roots, soil: horizon A, AE, and Bsv) was determined with the Kruskal–Wallis test, the median test, and the U Mann–Whitney test, at the level of significance of *p* = 0.05. For statistical calculations, the Excel spreadsheet and Statistica MS 2012 package were used.

## 5. Conclusions

A combination of acid and alkaline hydrolysis ensures the best extraction efficiency of insoluble-bound forms of polyphenols from plant and soil material.Alkaline re-hydrolysis resulted in an increase in the amount of the phenolics extracted from the plant material samples from 215% (hay) to 454% (roots), and from soil, from 55.7% (horizon Bsv) to 240% (horizon A) of their contents in acid hydrolysates.In the extracts from acid hydrolysates, in general, chlorogenic and *p*-hydroxybenzoic acids were dominant. In the alkaline extracts from the plant material, the highest content was recorded for *p-*coumaric and ferulic acids, and in the extracts from soil, for ferulic and chlorogenic acids.The values of the parameter V:S:C (the ratio of vanillyl to syringyl to cinnamyl compounds) recorded for the extracts from the plant material following the acid hydrolysis ranged from 1:1:2 to 1:1:4, and for the hydrolyses sum from 1:1:8 to 1:1:9. The values of the parameter V:S:C for soil, calculated for the hydrolyses in total, was 1:1:2–1:1:3, and following the acid hydrolysis only 1:1:1.5–1:1:2.It was demonstrated that the plant material of meadow soils, irrespective of the extraction method, showed a similar share of syringyl and vanillyl compounds and an advantage of cinnamyl compounds. In the humification process of meadow vegetation, the ratio of the share of V and S compounds did not change; however, the share of cinnamyl compounds in the V+S+C pool decreased.

## Figures and Tables

**Figure 1 molecules-25-05462-f001:**
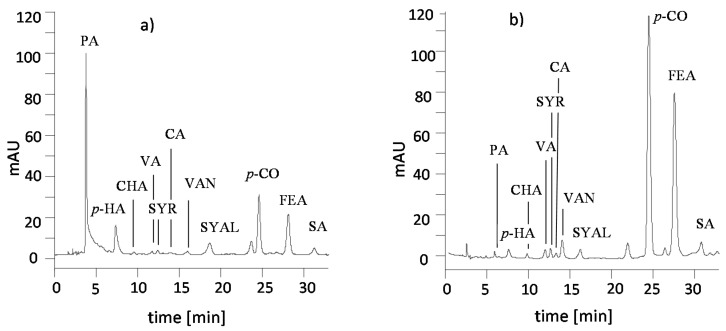
Selected RP-HPLC chromatogram of phenolic compounds of the extracts from hay: (**a**) following acid hydrolysis, (**b**) following alkaline re-hydrolysis.

**Figure 2 molecules-25-05462-f002:**
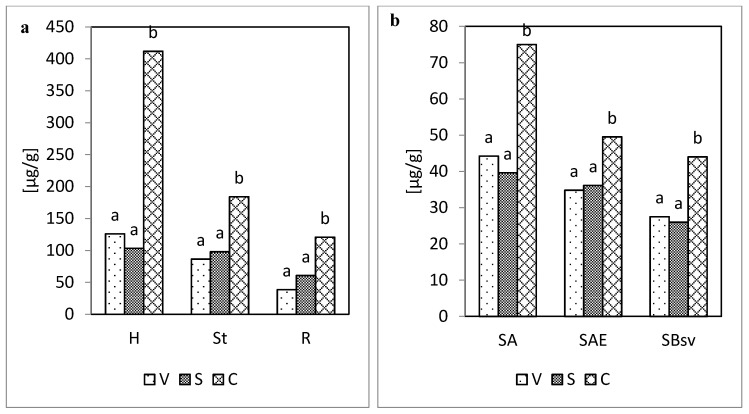
Content of vanillyl (V), syringyl (S), and cinnamyl (C) compounds following acid hydrolysis: (**a**) in the plant material (H, St, R), (**b**) in soil (SA, SAE, SBsv).

**Figure 3 molecules-25-05462-f003:**
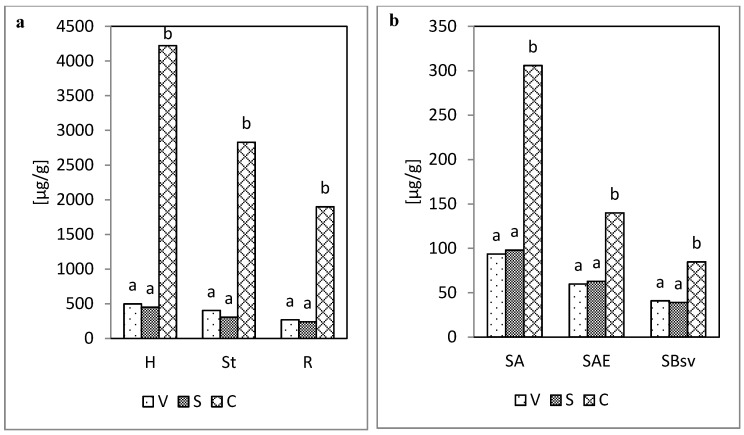
Content of vanillyl (V), syringyl (S), and cinnamyl (C) compounds following both hydrolyses (acid and alkaline): (**a**) in the plant material (H, St, R), (**b**) in soil (SA, SAE, SBsv).

**Figure 4 molecules-25-05462-f004:**
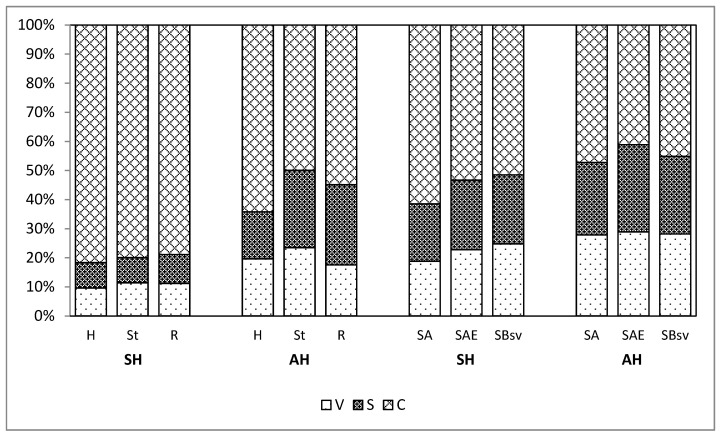
Share of vanillyl (V), syringyl (S), and cinnamyl (C) compounds in V+S+C, following acid hydrolysis (**AH**) and both hydrolyses (**SH**, acid and alkaline).

**Figure 5 molecules-25-05462-f005:**
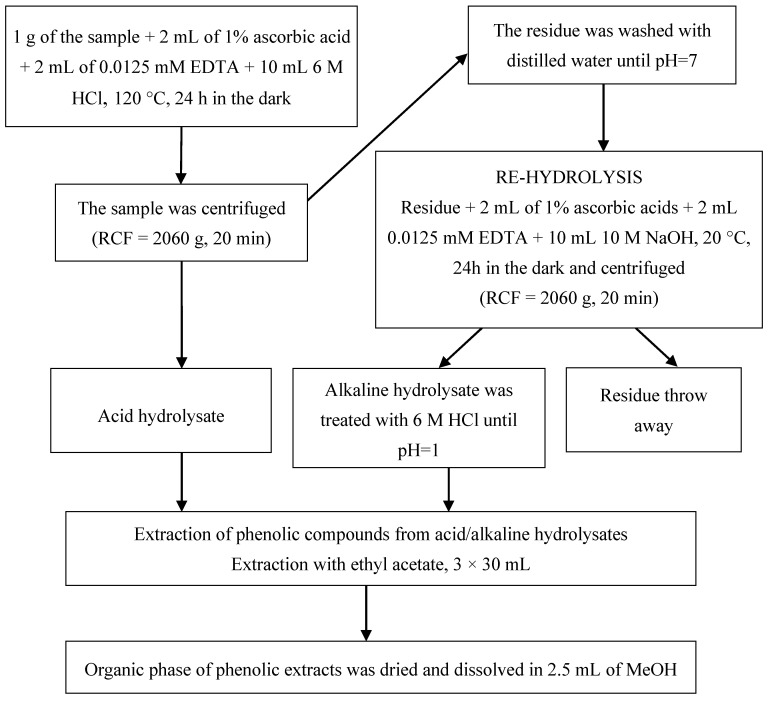
Extraction procedure for phenolic compounds from plant material and soil (modification of the methods described by Verma et al. 2009 and Ross et al. 2009).

**Figure 6 molecules-25-05462-f006:**
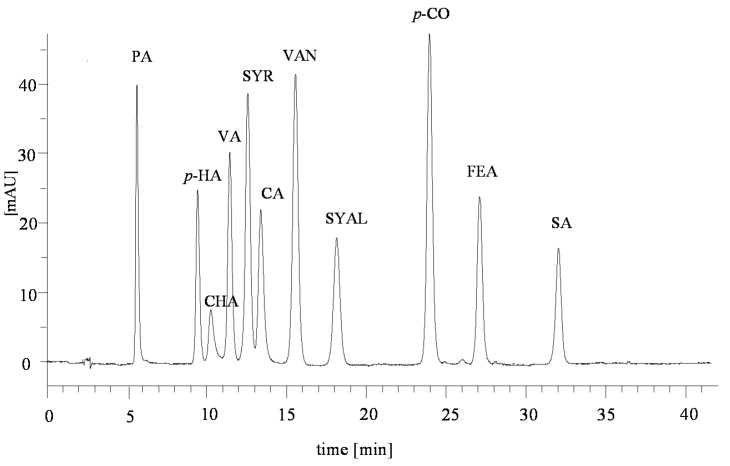
Phenolic compounds’ reference mixture chromatogram.

**Table 1 molecules-25-05462-t001:** Content of phenolic compounds in plant material (µg/g d.w.)

Symbol	Mean ± SD	Min	Max	SE	Mean ± SD	Min	Max	SE
	Acid hydrolysis				Alkaline Re-hydrolysis			
**Hay**								
PA	73.0 ± 62.8 ^a^	20.2	230.8	14.8	74.9 ± 19.6 ^a^	47.9	107.1	4.6
*p*-HA	72.2 ± 51.8 ^b^	13.8	182.5	12.2	59.9 ± 19.7 ^a^	19.4	79.3	4.7
CHA	710.8 ± 151.6 ^b^	530.0	947.4	35.7	130.9 ± 29.9 ^a^	90.0	183.1	7.0
VA	49.6 ± 17.2 ^a^	23.9	79.3	4.0	110.7 ± 24.2 ^b^	83.0	158.2	5.7
SYR	42.5 ± 21.2 ^a^	18.8	86.1	5.0	99.2 ± 29.5 ^b^	70.2	156.9	6.9
CA	64.2 ± 41.3 ^a^	13.4	147.8	9.7	166.6 ± 33.9 ^b^	105.5	228.3	8.0
VAN	76.4 ± 39.3 ^a^	24.2	149.7	9.3	261.4 ± 68.9 ^b^	179.5	385.6	16.2
SYAL	60.8 ± 32.4 ^a^	18.5	114.2	7.6	248.2 ± 48.9 ^b^	180.1	328.1	11.5
*p*-CO	145.9 ± 24.5 ^a^	105.9	189.7	5.8	1655.1 ± 304.7 ^b^	1275.0	2221.0	71.8
FEA	201.7 ± 41.7 ^a^	137.1	256.1	9.8	1988.0 ± 452.4 ^b^	1306.9	2500.4	106.6
SA	58.9 ± 32.6 ^a^	14.8	116.7	7.7	106.0 ± 61.2 ^b^	20.5	214.1	14.4
SUM	1556.1 ± 153.0 ^a^	1272.0	1816.7	36.1	4900.9 ± 685.7 ^b^	3900.2	6083.7	161.6
Sward								
PA	55.1 ± 26.1 ^a^	12.1	91.5	6.2	77.3 ± 20.2^b^	56.8	111.2	4.8
*p*-HA	90.6 ± 84.3 ^b^	18.3	278.4	19.9	78.6 ± 38.5^a^	33.6	143.8	9.1
CHA	315.3 ± 151.6 ^b^	126.5	638.6	35.7	142.6 ± 33.8^a^	100.5	202.5	8.0
VA	42.5 ± 32.5 ^a^	9.2	112.1	7.7	116.1 ± 24.4^b^	73.8	148.6	5.8
SYR	37.9 ± 23.5 ^a^	10.0	84.6	5.5	73.0 ± 20.1^b^	33.8	102.1	4.7
CA	77.9 ± 42.7 ^a^	36.9	164.8	10.1	127.4 ± 27.4^b^	77.0	165.1	6.5
VAN	44.0 ± 17.2 ^a^	17.3	73.8	4.1	201.2 ± 68.0^b^	117.9	348.1	16.0
SYAL	60.0 ± 28.0 ^a^	30.5	107.4	6.6	134.9 ± 53.6 ^b^	40.6	219.0	12.6
*p*-CO	47.1 ± 34.5 ^a^	7.3	107.3	8.1	1426.2 ± 253.5 ^b^	1005.8	1738.4	59.8
FEA	58.9 ± 30.1 ^a^	17.8	114.5	7.1	1092.2 ± 220.0 ^b^	900.6	1585.4	51.8
SA	12.8 ± 7.7 ^a^	2.8	25.7	1.8	129.0 ± 88.7 ^b^	42.8	263.2	20.9
SUM	842.2 ± 347.7 ^a^	507.0	1594.0	82.0	3598.5 ± 477.8 ^b^	3048.3	4592.5	112.6
Roots								
PA	33.6 ± 13.3 ^a^	10.9	56.1	3.1	71.6 ± 26.2 ^b^	22.5	105.9	6.2
*p*-HA	121.8 ± 13.7 ^b^	92.6	140.9	3.2	100.2 ± 38.4 ^a^	34.0	161.8	9.1
CHA	81.7 ± 41.9 ^a^	15.1	134.0	9.9	152.4 ± 30.4 ^b^	115.3	211.7	7.2
VA	16.2 ± 8.3 ^a^	3.3	30.4	2.0	118.2 ± 28.0 ^b^	57.2	149.8	6.6
SYR	20.6 ± 6.7 ^a^	9.7	32.0	1.6	82.1 ± 48.0 ^b^	26.0	164.0	11.3
CA	71.7 ± 38.1 ^a^	23.3	146.3	9.0	154.1 ± 38.9 ^b^	100.9	238.9	9.2
VAN	22.4 ± 9.7 ^a^	9.8	37.4	2.3	112.6 ± 15.0 ^b^	90.3	139.6	3.5
SYAL	40.0 ± 18.5 ^a^	10.9	71.4	4.4	96.7 ± 45.8 ^b^	20.0	165.6	10.8
*p*-CO	17.4 ± 13.9 ^a^	4.0	45.1	3.3	891.6 ± 214.2 ^b^	477.8	1128.6	50.5
FEA	31.6 ± 22.7 ^a^	3.4	65.7	5.3	732.5 ± 63.8 ^b^	629.2	843.7	15.0
SA	9.6 ± 2.0 ^a^	6.8	13.3	0.5	73.2 ± 21.0 ^b^	46.2	114.8	4.9
SUM	466.6 ± 81.8 ^a^	367.0	586.9	19.3	2585.2 ± 210.4 ^b^	2175.0	2883.9	49.6

^a,b^—Mean values with the same letter are not significantly different at 5%.

**Table 2 molecules-25-05462-t002:** Content of phenolic compounds in plant material and soil (µg/g d.w.).

Symbol	Mean ± SD	Min	Max	SE	Mean ± SD	Min	Max	SE
	Acid hydrolysis				Alkaline Re-hydrolysis			
**Horizon A**								
PA	7.6 ± 2.3 ^a^	3.6	11.8	0.5	24.1 ± 10.2 ^b^	8.1	44.0	2.4
*p*-HA	10.9 ± 5.3 ^a^	4.1	21.1	1.2	25.0 ± 15.7 ^b^	10.0	56.6	3.7
CHA	39.3 ± 30.3 ^a^	12.1	104.3	7.1	121 ± 62.2 ^b^	23.9	215.0	14.7
VA	26.4 ± 6.8 ^a^	17.8	40.2	1.6	30.2 ± 12.2 ^a^	10.5	50.9	2.9
SYR	13.7 ± 4.0 ^a^	9.8	23.9	0.9	26.2 ± 9.7 ^b^	10.4	42.8	2.3
CA	42.0 ± 21.8 ^a^	23.6	90.1	5.1	33.4 ± 13.6 ^a^	10.3	54.0	3.2
VAN	17.8 ± 12.0 ^a^	6.0	43.7	2.8	19.1 ± 10.7 ^a^	9.0	40.4	2.5
SYAL	25.9 ± 11.5 ^a^	8.1	49.9	2.7	32.1 ± 14.9 ^a^	10.6	57.1	3.5
*p*-CO	3.1 ± 2.6 ^a^	0.8	8.7	0.6	84.1 ± 42.8 ^b^	15.1	144.3	10.1
FEA	29.9 ± 9.0 ^a^	17.2	46.0	2.1	113.3 ± 46.4 ^b^	30.3	177.6	10.9
SA	4.5 ± 3.4 ^a^	2.0	13.5	0.8	23.2 ± 16.2 ^b^	4.6	49.3	3.8
SUM	221.2 ± 44.5 ^a^	166.5	307.6	10.5	531.6 ± 132.3 ^b^	334.9	770.9	31.2
	Horizon AE							
PA	16.3 ± 7.7 ^b^	6.8	30.4	1.8	7.2 ± 4.2 ^a^	3.0	17.9	1.0
*p*-HA	77.0 ± 27.8 ^b^	40.4	126.1	6.6	10.9 ± 4.7 ^a^	5.1	21.3	1.1
CHA	54.1 ± 24.6 ^a^	21.2	90.5	5.8	49.5 ± 24.8 ^a^	10.0	79.0	5.8
VA	17.3 ± 9.3 ^b^	6.1	34.0	2.2	12.0 ± 8.6 ^a^	3.2	29.4	2.0
SYR	16.4 ± 14.0 ^a^	2.0	43.5	3.3	13.1 ± 5.5 ^a^	3.3	23.1	1.3
CA	35.8 ± 19.9 ^a^	6.1	66.2	4.7	28.8 ± 16.0 ^a^	7.2	55.2	3.8
VAN	17.5 ± 11.3 ^a^	3.2	34.5	2.7	12.9 ± 6.6 ^a^	4.0	27.0	1.6
SYAL	19.7 ± 12.3 ^b^	9.2	47.2	2.9	13.6 ± 12.5 ^a^	3.6	44.4	2.9
*p*-CO	4.3 ± 4.0 ^a^	0.4	13.9	0.9	11.4 ± 7.7 ^b^	4.7	29.1	1.8
FEA	9.4 ± 3.7 ^a^	2.1	13.4	0.9	50.2 ± 14.1 ^b^	30.2	77.3	3.3
SA	5.4 ± 3.9 ^a^	1.8	15.9	0.9	12.3 ± 6.8 ^b^	6.3	28.8	1.6
SUM	273.1 ± 50.2 ^b^	187.2	355.7	11.8	221.9 ± 47.9 ^a^	148.7	296.0	11.3
	Horizon Bsv							
PA	13.3 ± 7.6 ^b^	3.3	29.9	1.8	8.3 ±2.6 ^a^	5.3	13.4	0.6
*p*-HA	57.5 ± 29.2 ^b^	12.9	92.3	6.9	12.9 ± 5.4 ^a^	5.1	24.8	1.3
CHA	44.4 ± 21.4 ^b^	18.2	86.7	5.0	21.6 ± 7.4 ^a^	11.1	35.0	1.7
VA	14.6 ± 9.0 ^b^	6.3	34.7	2.1	8.7 ± 6.1 ^a^	4.0	25.0	1.4
SYR	12.6 ± 3.2 ^b^	6.4	17.2	0.7	6.4 ± 3.7 ^a^	2.0	13.8	0.9
CA	27.5 ± 18.9 ^b^	8.4	60.0	4.5	8.6 ± 4.6 ^a^	3.3	17.2	1.1
VAN	12.9 ± 6.1 ^b^	6.5	24.4	1.4	4.6 ± 1.6 ^a^	2.4	8.0	0.4
SYAL	13.4 ± 15.1 ^b^	1.6	48.6	3.6	6.6 ± 1.5 ^a^	3.7	9.6	0.3
*p*-CO	4.3 ± 2.4 ^a^	1.4	9.3	0.6	9.5 ± 6.8 ^b^	2.0	22.7	1.6
FEA	12.2 ± 8.6 ^a^	3.3	30.7	2.0	22.6 ± 15.3 ^b^	7.8	58.7	3.6
SA	3.6 ± 2.2 ^a^	1.1	6.8	0.5	10.8 ± 9.2 ^b^	3.0	33.0	2.2
SUM	216.5 ± 30.2 ^b^	154.5	262.6	7.1	120.5 ± 29.7 ^a^	75.8	184.1	7.0

^a,b^—Mean values with the same letter are not significantly different at 5%.

**Table 3 molecules-25-05462-t003:** Ratio of the share of vanillyl (V), syringyl (S), and cinnamyl (C) compounds V:S:C in hydrolysates of plant material (H, St, R) and soil (SA, SAE, SBsv).

Material	Sum of Hydrolyses	Acids Hydrolysis
H	1:1:9	1:1:4
St	1:1:9	1:1:2
R	1:1:8	1:1.5:3
SA	1:1:3	1:1:2
SAE	1:1:2.5	1:1:1,5
SBsv	1:1:2	1:1:1.5

**Table 4 molecules-25-05462-t004:** Basic parameters of soils and grain size composition (mm).

Horizon		TOC	Nt	pH	0.2-0.05	0.05–0.002	<0.002
		(g/kg)	(g/kg)		(%)		
A	Mean	39.72	3.65	-	91.74	8.25	n.d. **
	Min	30.20	2.80	5.2	89.45	6.68	-
	Max	45.20	4.15	6.5	93.32	10.53	-
	SE *	6.00	0.49	-	1.29	1.28	-
AE	Mean	28.73	2.77	-	91.33	9.20	n.d.
	Min	21.40	2.13	5.6	88.70	6.19	-
	Max	35.60	3.41	6.7	93.81	11.82	-
	SE	6.31	0.57	-	1.83	2.21	-
Bsv	Mean	15.83	1.44	-	89.30	10.20	0.02
	Min	10.00	0.47	5.9	85.72	8.58	0.02
	Max	24.00	2.49	6.7	91.42	13.28	0.02
	SE	6.63	0.84	-	2.19	1.90	-

* standard error, ** not determined, TOC—Total organic carbon, Nt—Total nitrogen.

**Table 5 molecules-25-05462-t005:** Regression coefficient, LOD, and LOQ of phenolic compounds reference.

Name of the Standard	Regression Coefficient R^2^	LOD μg/mL	LOQ μg/mL
Protocatechuic acid	0.9831	0.22	0.67
*p*-Hydroxybenzoic acid	0.9801	0.41	1.23
Chlorogenic acid	0.9806	0.24	0.71
Vanillic acid	0.9836	0.25	0.75
Syringic acid	0.9863	0.35	1.04
Caffeic acid	0.9817	0.26	0.78
Vanillin	0.9865	0.17	0.51
Syringaldehyde	0.9818	0.15	0.45
*p*-Coumaric acid	0.9875	0.30	0.91
Ferulic acid	0.9858	0.23	0.70
Salicylic acid	0.9802	0.24	0.73

LOD—Limit of detection, LOQ—Limit of quantification.

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
