# Peer review of "Content of Phenolic Compounds in Meadow Vegetation and Soil Depending on the Isolation Method"

_molecules, 2020, doi:10.3390/molecules25225462_

Round 1
Reviewer 1 Report
In addition to the comments in the pdf, I would strongly advise the authors to include the novelty of their work, the significance, and why it is interesting to the audience.
Important aspects are missing - such as the limitations of the study - and these are key to a certain level of self-criticism neccessary in science.

Author Response
Dear Reviewer,
The paper's Authors wish to thank for all the precious comments and guidelines; we have made an attempt at considering all of them.
I hope that the corrections introduced are satisfactory. All the changes have been marked in the text.
With best regards,
Magdalena Banach-Szott
Line 2- 4: the topic has been inconsiderably changed:
There was: Content of phenolic compounds in meadow vegetation and soil depending on the selection of the isolation method
There is: Content of phenolic compounds in meadow vegetation and soil depending of the isolation method
Line 10-12:
There was: The aim of this paper has been to determine the effect of the extraction method on the amount of phenolic compounds (PC) in the plant material in soil and, as a consequence, on the parameters used e.g. to determine the degree of lignin transformation in soils.
There is: The aim of this paper was to determine the effect of the hydrolysis method on the amount of phenolic compounds in the plant material in soil and, as a consequence, on the parameters to determine the degree of lignins transformation in soils.
Line 16-21: changed
There was: The quantitative and qualitative analysis of phenolic compounds was made with High Performance Liquid Chromatography. There has been, that the content of phenolic compounds in the extracts depend on the method and the share of the method in the amount of phenolic compounds acquired is determined by the type of the research materials. The amount of phenolic compounds contained in the alkaline hydrolysates accounted for 55.7 (soil, horizon Bsv) – 45.4% (roots) of their content in acid hydrolysates. In the extracts from acid hydrolysates chlorogenic and p-hydroxybenzoic acids were dominant.
There is: The quantitative and qualitative analysis of phenolic compounds was performed with High Performance Liquid Chromatography with the DAD detector. The content of phenolic compounds in the extracts depended on the hydrolysis method and it was determined by the type of the research material. The amount of phenolic compounds contained in the alkaline hydrolysates accounted for 55.7 (soil, horizon Bsv) – 454% (roots) of their content in acid hydrolysates. In the extracts from acid hydrolysates, chlorogenic and p-hydroxybenzoic acids were dominant.
Line 30-35: changed
There was: As for the chemical composition, phenolic acids, as lignin derivatives, present in plant cells, can be divided into groups which differ in the number of and the place of substitution of hydroxyl and methoxyl groups:
–hydroxybenzoic acid and its derivatives, e.g. dihydroxybenzoic, protocatechuic, salicylic, syringic, vanillic acids; hydroxycinnamic acid
–and its derivatives, e.g. cinnamyl, ferulic, p-coumaric, caffeic acids.
There is: As for the chemical composition, phenolic acids, as lignin derivatives, present in plant cells, can be divided into groups which differ in the number of and the place of substitution of hydroxyl and methoxyl groups:
–hydroxybenzoic acid and its derivatives, e.g. dihydroxybenzoic, protocatechuic (PA), salicylic (SA), syringic (SYR), vanillic acids (VA);
– hydroxycinnamic acid and its derivatives, e.g. cinnamic, ferulic (FEA), p-coumaric (p-CO), caffeic acids (CA).
Line 40-41: changed:
There was: … extraction, extraction combined with hydrolysis, ultrasound or microwave-radiation-assisted extraction. Very often the simple extraction of phenolic compounds shows a low performance as it mostly ….
There is: …extraction combined with hydrolysis, ultrasound or microwave-radiation-assisted extraction [1, 3-4]. Very often the simple extraction of phenolic compounds using organic solvents shows a low performance as it mostly …..
Line 49-50: changed:
There was: …produced from the alkaline hydrolysis, whereas,….
There is: ….. produced from the acid hydrolysis, whereas ….
line 51-54:
There was: Besides, the authors show that an alkaline hydrolysis (the so-called re-hydrolysis) after acid hydrolysis contributes to the contents of phenolic compounds often bigger than as a result of acid hydrolysis, which demonstrates that some phenolic compounds are labile in the acid environment and some – in the alkaline one.
There is: Besides, the authors show that some phenolic compounds are soluble in the acid environment and some – in the alkaline one.
Line 56-58:
There was: …..who compare three sample hydrolysis methods: alkaline hydrolysis 10 M NaOH, followed by acid hydrolysis using concentrated HCl; alkaline hydrolysis alone 2 M NaOH and acid hydrolysis alone using 12 M HCl
There is: They compare three sample hydrolysis methods: alkaline hydrolysis followed by acid hydrolysis, alkaline hydrolysis alone and acid hydrolysis alone.
Line 64-65:
There was: … both types of hydrolysis should be applied. An additional factor enhancing the isolation of phenolic compounds is adding AA and EDTA.
There is: … both types of hydrolysis should be applied and AA and EDTA should be added.
Line 95 (Figure 1). The figure provides the explanation which peak corresponds to which compound.
Line 98-101: The sentence has been simplified.
There was: The quantitative analysis of the chemical composition of the extracts from plant material hydrolysates sampled 5 and 25 m away from the irrigation ditch has shown that in the extracts from re-hydrolysates the significantly higher contents of the total phenolic compounds analysed were recorded as compared with their contents in the extracts from acid hydrolysates (Table 2).
There is: The quantitative analysis of the phenolics of the hydrolysates from the plant material has shown that in the extracts from alkaline re-hydrolysates the significantly higher contents of the total phenolic compounds analyzed were recorded, as compared with their contents in the extracts from acid hydrolysates (Table 1)
Line 104-105: the word “significantly” has been removed. The sentence refers to all the plant material samples discussed in this paragraph.
Line 114-131: We agree with the Reviewer that some data quoted in this fragment can be found in the table. The authors of the paper have done it on purpose to stress some dependencies and to refer them to the percentage share (the values of which are cited in the text only).
Line 173-175: Obviously there are literature reports on the application of alkaline hydrolysis after acid hydrolysis, however, mostly in reference to plant material. There is no data referring to the soil samples and there are indeed no reports in which the method is used for the plant and soil at the same time.
Line 180: re-hydrolysis – removed
Line 188-191: In the text the authors have tried to apply at the same time the full names and abbreviations for the reader to find the data in the tables more easily.
Line 211: The sentence “It is unacceptable to apply the acid hydrolysis only for the plant material samples.” has been removed.
Line 266: The coordinates have been added.
Line 283:
There was: acid hydrolysis involving the treatment of the sample with HCl solution;
There is: acid hydrolysis with a HCl solution
Line 286: The course of the process of acid hydrolysis and re-hydrolysis is presented in Fig. 5.
Line 285-288: They have been removed as the fragment has been a repetition of the information provided in Figure 5.
Figure 5. The arrow is correctly directed. After centrifugation, acid hydrolysate and sediment were produced. The sediment was exposed to alkaline hydrolysis and not to acid hydrolysis.
Line 302-304
There was: The samples for analysis were prepared by combining 0.5 mL filtered applying the nylon injection filter (φ=0.45 µm) of the extract of phenolic compounds with 0.5 mL of eluent A. The injection volume was 10 µL. A gradient separation program at the flow rate of 1.3 mL/min.
There is: The samples for analysis were prepared by combining 0.5 mL filtered (nylon filter, φ=0.45 µm) of the extract of phenolic compounds with 0.5 mL of eluent A. The injection volume was 10 µL (three replications). The gradient division program was applied at the rate flow of 1.3 mL/min.
Line 308-312
There was: Phenolic compounds were identified from the pattern of the chromatogram for the reference solution of phenolic compounds the list of which is provided in Table 1. The content of phenolic compounds is given in µg/g d. w. The quantitative analysis of the phenolic compounds identified was performed using the reference curves of the dependence of the peak area for the concentration of phenolic compound (μg/mL).
There is: The pattern of the chromatogram for the reference solution of phenolic compounds and their retention times (Figure 6) facilitated identifying the following phenolic compounds: protocatechuic acid (PA), p-hydroxybenzoic acid (p-HA), chlorogenic acid (CHA), vanillic acid (VA), syringic acid (SYR), caffeic acid (CA), vanillin (VAN), syringaldehyde (SYAL), p-coumaric acid (p-CO), ferulic acid (FEA) and salicylic acid (SA) (Table S1). The content of phenolic compounds is given in µg/g d. w. The quantitative analysis of the phenolic compounds identified was performed using the external reference curves of the dependence of the peak area for the concentration of phenolic compound (μg/mL). The reference response curve for each phenolic compound was a linear regression with correlation coefficients > 0.98.
Line 320-343. The authors of the paper have not abandoned the conclusions but the conclusions have been changed considerably, leaving just the most essential results of the paper.
Reviewer 2 Report
The manuscript with the title ‘Content of phenolic compounds in meadow vegetation and soil depending on the selection of the isolation method’ is interesting and fits the scope of Molecules journal. However, I found several weak points that should be addressed before considering for publication. My comments and concerns are listed below, in the hope they will help to improve the scientific quality of this manuscript.
GENERAL COMMENTS:
- Methods of extracting phenolic acids from various plant samples using acid and alkaline hydrolysis have already been described in the literature, the authors do not propose major modifications, and this slightly reduces the novelty of their research.
- The authors compare the extraction efficiency of insoluble forms of polyphenols using two methods - acid vs alkaline hydrolysis, but for the alkaline hydrolysis they used samples previously subjected to acid hydrolysis, and I wonder if they should not provide the same initial conditions for both procedures (fresh samples)? Moreover, I wonder if the authors should not take into account and compare the results of 3 methods for a complete picture of the effect of the type of hydrolysis on the extraction of polyphenols - acid hydrolysis vs alkaline hydrolysis vs double (acid + alkaline) hydrolysis?
- In my opinion, the authors are not fully legitimate to generalize that lignin is the only source of simple polyphenols detected in plant and soil samples, forgetting that some of them can also be found in a free and soluble state. Therefore, I wonder if it was not necessary to get rid of these free forms of polyphenols by extraction with an organic solvent prior to sample hydrolysis – the liberation of phenolics from insoluble-bound forms?
- I suggest the revision of the manuscript by a native speaker.
- All abbreviations in the text, tables, and figures should be explained in their first appearance, such as d.w., H, St, R, TOC, Nt, etc.
ABSTRACT:
- lines 13-14: ‘soils (horizon A, AE and Bsv, Albic Brunic Arenosol).’, here should be indicated that Albic Brunic Arenosol is a soil type but not a horizon.
- lines 16-19: ‘There has been, that the content of phenolic compounds in the extracts depend on the method and the share of the method in the amount of phenolic compounds acquired is determined by the type of the research materials.’, the sentence is not very clear, consider re-writing it.
- The abstract lacks a conclusion of the research, such as – A combination of acid and alkaline hydrolysis ensures the best extraction efficiency of insoluble-bound forms of polyphenols from plant and soil material.
INTRODUCTION:
- line 27: ‘polyphenolic compounds’, consider changing to ‘polyphenols’.
- line 34: move ‘hydroxycinnamic acid’ to next line 35
- line 35: ‘cinnamyl’, change to cinnamic?
- lines 36-38: ‘Another group of phenolic compounds are depsides, which are combination of two or more molecules of phenolic acids, e.g. chlorogenic acid is produced as a result of the combination of the carboxyl group of caffeic acid with phenolic group of quinic acid [2].’, according to the definition of a depside – two or more linked phenolic acids, given by the authors, can chlorogenic acid be classified as depside? Quinic acid does not have a phenolic structure and phenolic group. Please check it.
- line 41: ‘Very often the simple extraction of phenolic compounds shows a low performance…’, consider changing to ‘Very often the simple extraction of phenolic compounds using organic solvents shows a low performance’.
- lines 42-43: consider changing ‘bound forms’ to ‘insoluble-bound forms’, because some bound forms of phenolic compounds, such as glycosides, can be soluble and extracted with aqueous organic solvents.
- line 50: ‘produced from the alkaline hydrolysis’, please check and possibly change to ‘acid hydrolysis’.
RESULTS:
- Table 1: I have some doubts about whether the inclusion of such a large table, showing the structures of simple and known polyphenols, is justified. I think it can be removed/transferred to the supplementary materials. Instead of Table 1, the authors may, for example, place a pictorial drawing showing the chromatograms of all types of plant and soil samples analyzed, along with marked identified polyphenols.
- Authors often use the wording ‘acid hydrolysis and re-hydrolysis’, for the sake of clarity, I would recommend using the phrase ‘acid hydrolysis and alkaline re-hydrolysis’ more frequently.
- Figure 1: mV? please check the unit, if it should not be mAU (adequate for DAD/PDA detectors)?
- Tables 2 and 3: µg/g s.m.?
DISCUSSION:
- lines 212-219: ‘Isolating and identifying phenolic compounds allow for dividing those compounds into 3 groups referred to as: –vanillyl compounds (V), as the total content of vanillin and vanillic acid derived from coniferyl alcohol; –syringyl compounds (S), as the total content of syringaldehyde and syringic acid, derived from sinapyl alcohol; –cinnamyl compounds (C), as the total content of ferulic acid, p-coumaric acid and caffeic acid, derived from coumaryl alcohol.’, the authors conclude that the total amounts of aforementioned phenolic compounds present in plant and soil samples are derived from three monolignols, but what about the free and soluble forms of these compounds that are also present in these samples?
- lines 235-238: ‘As seen from the dependencies presented in Figure 2 and 3, irrespective of the methods of assaying of the phenolic compounds (acid hydrolysis, sum of both hydrolyses) applied, ‘no significant differences in the content of vanillyl and syringyl compounds were demonstrated’, and cinnamyl compounds were dominant.’, I think it should be specified between which samples there were no differences found because someone can conclude that there were no differences in the levels of V, S and C compounds between for example hay and soil-Bsv.
- Figures 2 and 3: I wonder how the statistical ANOVA comparison for samples was performed. Have all plant and soil samples been compared, for example, hay vs. sward vs. roots? For example, in Fig 2A all bars for C compounds have assigned the same letter and the bars differ significantly in height. Please, check it.
MATERIALS AND METHODS:
- Lack of Chemicals section, and yet different reagents were used for the research.
- Materials section: there is no information provided when the samples were taken (date of collection), what size of samples were taken, how the samples were secured (drying?) and prepared (milling?) for analysis. The samples were collected 5 and 25 m away from the irrigation ditch, were they later mixed? Information about what and how many samples were taken could be given more clearly, I understand that there were 3 stands (x 2 distances), thus 6 independent samples were taken for each type of plant (hay, sward, and roots) and soil (A, AE and Bsv horizons) material.
- Methods section: there is no information provided if the method of extraction was based on previous research (appropriate citation) or was designed by authors. No hydrolysis parameters were given: the amount of sample, duration, temperature, whether in the dark, etc.?
- Figure 5: In my opinion, the scheme is not very clear, I would separate both procedures – acid and alkaline hydrolysis. According to the scheme, the first reagent added to the sample was HCl or NaOH, and later the antioxidant was added, is it correct? Was the sample also centrifuged after the alkaline hydrolysis? After alkaline hydrolysis, the hydrolyzate was treated with HCl, what was the purpose of this step? Finally, a 90 ml of extract was collected (very diluted), was it further concentrated/dried and dissolved in lower volume?
- Qualitative and quantitative analysis: Please, provide necessary information for equipment used (producer, country). There is no information provided about the origin and purity of reference standards, characteristic of calibration curves (linearity, R2, LOD, LOQ), number of injections (analysis) per sample.
- lines 307-308: ‘The detection was made at the wavelength of 280 nm.’, why this wavelength was chosen; the analyzed phenolic substances have different UV absorption maxima, wouldn't it be more methodically correct to quantify at the maximum for each compound?
- Statistical analysis: Consider changing ‘Duncan test’ to ‘ANOVA test followed by Duncan’s multiple comparison test’.
Author Response
Dear Reviewer,
The paper's Authors wish to thank for all the precious comments and guidelines; we have made an attempt at considering all of them.
I hope that the corrections introduced are satisfactory. All the changes have been marked in the text.
With best regards,
Magdalena Banach-Szott
GENERAL COMMENTS:
The authors of the paper, considering the literature reports (e.g., Verma et. al. 2009) pointing to an inconsiderable share of free phenolics assaying with the HPLC-UV method in the total pool of phenolics, have abandoned that assay. The key objective of the research has been to determine:
- to what extent the application of alkaline hydrolysis after acid hydrolysis increases the amount of the phenolics produced.
- and whether the type of the material (plant material, soil) has an impact on the efficiency of alkaline extraction applied following the acid hydrolysis (no such comparisons have been found in literature).
Since more frequently the Authors of the paper have come across the papers in which the authors based only on acid hydrolysis, and the Authors in the earlier research also used the acid hydrolysis, this time the Authors decided to determine to what extent the method is efficient.
As already reported earlier, the amounts of free phenols are inconsiderable and, as compared to soils, also inconsiderable and so this assay has been disregarded.
The manuscript has been proofread by a native speaker.
All the abbreviations are explained whenever they appear for the first time.
ABSTRACT:
lines 13-14
there was: The research involved the plant material samples (hay, sward and roots) and soils (horizon A, AE and Bsv, Albic Brunic Arenosol)
there is: The research involved the plant material samples (hay, sward and roots) and soils - Albic Brunic Arenosol (horizon A, AE and Bsv)
lines 16-19
there was: There has been, that the content of phenolic compounds in the extracts depend on the method and the share of the method in the amount of phenolic compounds acquired is determined by the type of the research materials.
there is: The content of phenolic compounds in the extracts depended on the hydrolysis method and it was determined by the type of the research material.
In the abstract, compliant with the suggestion by the Reviewer, a conclusion of the research has been provided:
A combination of acid and alkaline hydrolysis ensures the best extraction efficiency of insoluble-bound forms of polyphenols from plant and soil material.
INTRODUCTION:
line 27
‘polyphenolic compounds’, has been changed to ‘polyphenols’.
line 34
‘hydroxycinnamic acid’ has been moved to line 35
line 35
‘cinnamyl’ has been changed to cinnamic
lines 36-38
there was: Another group of phenolic compounds are depsides, which are combination of two or more molecules of phenolic acids, e.g. chlorogenic acid is produced as a result of the combination of the carboxyl group of caffeic acid with phenolic group of quinic acid [2].
there is: Another group of phenolic compounds are depsides the molecule core of which includes an ester bond, e.g., chlorogenic acid (CHA) is produced as a result of the combination of the carboxyl group of caffeic acid with phenolic group of quinic acid [2].
line 41
there was: Very often the simple extraction of phenolic compounds shows a low performance…
There is: Very often a simple extraction of phenolics using organic solvents shows a low performance…
lines 42-43
there was: To isolate the bound forms, it becomes justifiable to apply preliminary hydrolysis, followed by extraction [1,3]
There is: To isolate the insoluble-bound forms, it becomes justifiable to apply preliminary hydrolysis, followed by extraction [1,3]
line 50
There was: Verma et al. [4] have demonstrated that higher contents of p-coumaric, vanillic and sinapic acids were recorded as a result of the alkaline hydrolysis, as compared with the contents of the compounds produced from the alkaline hydrolysis.
There is: Verma et al. [4] have demonstrated that higher contents of p-coumaric, vanillic and sinapic acids were recorded as a result of the alkaline hydrolysis, as compared with the contents of the compounds produced from the acid hydrolysis.
RESULTS
Table 1
Table 1 has been transferred to the supplementary materials. As a consequence, the table numbering has been changed.
As a result of the present research, 36 chromatograms (without replications) have been received. According to the Authors, providing them in the paper (even as supplement) will not contribute with any essential information. The paper provides example differences noted in the pattern of chromatograms on the peak intensity, while the chromatogram has been added to a mixture of references.
In section 4.2.2. (currently 4.3.2) Qualitative and quantitative analysis of extracts containing phenolic compounds (Methods) a fragment of the compounds identified has been provided.
The pattern of the chromatogram for the reference solution of phenolic compounds and their retention times (Figure 6) facilitated identifying the following phenolic compounds: protocatechuic acid (PA), p-hydroxybenzoic acid (p-HA), chlorogenic acid (CHA), vanillic acid (VA), syringic acid (SYR), caffeic acid (CA), vanillin (VAN), syringaldehyde (SYAL), coumaric acid (p-CO), ferulic acid (FEA) and salicylic acid (SA) (Table S1).
As suggested by the Reviewer, the wording “re-hydrolysis” has been replaced with the wording “alkaline re-hydrolysis”
The Authors agree with the Reviewer that the correct unit is mAU and it has been corrected in Figure 1.
In Tables 2 and 3, currently Tables 1 and 2, the unit provided is correct, namely: µg/g s.m.
DISCUSSION:
lines 212-219: ‘The Authors of the paper in the introduction of the Response to reviews account for a lack of the results on free phenolics
lines 235-238: The fragment concerning the description to Figures 2 and 3 (currently Figures 1 and 2) has been edited and now it reads as follows:
Irrespective of the methods of assaying the phenolic compounds (acid hydrolysis, sum of both hydrolyses) applied, the contents of vanillyl, syringyl and cinnamyl compounds were definitely higher in the plant material, as compared with their contents in soil. Interestingly, both for the plant material samples and for the soil samples, no significant differences in the content of vanillyl and syringyl compounds were found, whereas cinnamyl compounds were dominant.
Figures 2 and 3
The Authors have corrected the description of the statistics methods used to compare Figures 2 and 3 (currently Figures 1 and 2)
There was: The results were statistically verified by determining the following: mean, maximum and minimum, standard deviation (SD) and standard error (SE). The differences of the values of the parameters depending on the extraction method determined with the Duncan test, at the level of significance of p=0.05. The above relationships were determined using statistical software STATISTICA MS 2012.
There is: The results were statistically verified by determining the following: mean, maximum and minimum, standard deviation (SD) and standard error (SE). The significance of the differences between the content of vanillyl, syringyl and cinnamyl compounds for each variant (hay, sward, roots, soil: horizon A, AE and Bsv) was determined with the Kruskal-Wallis test, the median test and the U Mann-Whitney test, at the level of significance of p=0.05. For statistical calculations, the Excel spreadsheet and Statistica MS 2012 package were applied.
MATERIALS AND METHODS:
Chemicals section has been added and it reads as follows:
4.1. Chemicals
HCl, NaOH, ascorbic acid, EDTA, ethyl acetate: p.a., provided by Avantor Performance Materials, Poland. Phenolic compounds, analytical standard: protocatechuic acid, p-hydroxybenzoic acid, chlorogenic acid, caffeic acid, vanillin, ferulic acid, salicylic acid provided by Sigma-Aldrich; vanillic acid, syringic acid, Syringaldehyde, coumaric acid provided by Fluka
Acetonitrile, methanol, acetic acid, HPLC purity, provided by Avantor Performance Materials, Poland.
The water was purified using the Millipore Milli-Q system.
Materials section: The missing data on the sample preparation method and the number of the samples studied have been added:
There was: The samples were taken in the country’s and Europe’s unique perennial grasslands complex, the Czerskie Meadows. The Czerskie Meadows Complex was established in the mid 19th century and it is located in various parts of the Tuchola Forest (Poland). In total it covers the area of about 2000 ha. The uniqueness of that complex is related to the irrigation system, namely the slope-and-flood system applied back then, which, in most areas of the Czerskie Meadows, has still survived.
The samples were taken from the plot where irrigation had been stopped since 1996, referred to by the founders as the Green Meadow. Plant material (hay, sward and roots) and soil (horizons A, AE and Bsv) were sampled from three specified stands. At each stand the soil was sampled 5 and 25 m away from the irrigation ditch. In total soil was sampled from 6 soil profiles (a total of 18 plant material and 18 soil samples). Soil in all the profiles was analysed compliant with the WRB systematics to Brunic Albic Arenosol. The basic soil properties are presented in Table 5 [21]
There is: The samples were taken from the plot where irrigation had been stopped since 1996, referred to by the founders as the Green Meadow. Plant material (hay, sward and roots) and soil (horizons A, AE and Bsv) were sampled from three specified stands. At each stand the soil was sampled 5 and 25 m away from the irrigation ditch. In total soil was sampled from 6 soil profiles (a total of 18 plant material and 18 soil samples).
The samples were taken from the plot where irrigation had been stopped since 1996, referred to by the founders as the Green Meadow. The plant material (hay, sward and roots) and soil (horizons A, AE and Bsv) were sampled from three specified stands. At each stand the soil was sampled 5 and 25 m away from the irrigation ditch. In total, soil was sampled from 6 soil profiles (a total of 18 plant material and 18 soil samples). The soil samples were dried in room temperature and sieved (2 mm). The plant material samples were dried in room temperature and ground. There were analysed 18 plant material samples and 18 soil samples, whereas the results provided in the tables are mean values for the three stands specified and two distances (5 and 25 m). The soil in all the profiles was analysed in compliance with the WRB systematics as Brunic Albic Arenosol. The basic soil properties are presented in Table 4 [21].
The missing data on the extraction procedure has been supplemented by adding the information in Figure 5. The figure heading has been supplemented by adding the literature items based on which the method of hydrolysis and phenolics extraction applied in this paper was developed.
Figure 5 has been supplemented with the missing data. The Authors of the paper have not divided that figure into two as:
- the process of hydrolysis is made for the same sample,
- the stage of phenolics extraction is the same for both hydrolysates.
After the alkaline hydrolysis, the hydrolysate was treated with HCl to increase the efficiency of extraction with ethyl acetate.
Qualitative and quantitative analysis: The data has been supplemented
lines 307-308: Since the identification of phenolics was performed based on a mixture of references, the wavelength applied was adequate for the best separation and for the intensity of each compound studied in the mixture.
Statistical analysis: The methodology has been provided in Chapter 4.3 (currently 4.4) Statistical analysis has been described by mistake. The description of the statistics methods has been changed.
Round 2
Reviewer 1 Report
I encourage again the authors to include two things in the manuscript: the limitations of the work and possible suggestions for future applications (as this links to the relevance of their work in the context of the research field).
There are still some minor corrections - marked in the pdf included.

Author Response
Dear Reviewer,
The paper's Authors wish to thank for all the precious comments and guidelines.
I hope that the corrections introduced are satisfactory. All the changes have been marked in the text.
With best regards,
Magdalena Banach-Szott
The Authors of the paper address their research to all the researchers of the transformation of organic matter in the soils of various ecosystems. The results point mostly to the need of applying acid hydrolysis and alkaline re-hydrolysis to provide the most accurate information on the content of phenolic compounds in soils (conclusion 1).
The research performed in the plant-soil system has pointed to a possibility of using the V:S:C parameter calculated from the content of phenolic compounds contained in acid hydrolysates and re-hydrolysates produced from soil samples to determine the level of transformation of organic matter in soils as well as the origin of plant materials undergoing the transformation (conclusions 4,5).
Some limitation of the method can be that its time-consuming; assaying the content of phenolic compounds based on acid hydrolysis alone or alkaline hydrolysis alone shortens the time considerably.
Line 13- 15:
There was: Phenolic compounds were at two stages by applying the acidic hydrolysis followed by alkaline re-hydrolysis.
There is: Phenolic compounds were isolated at two stages by applying acidic hydrolysis followed by alkaline re-hydrolysis.
Line 16:
There was: …High Performance Liquid Chromatography with the DAD detector.
There is: …High Performance Liquid Chromatography with a DAD detector.
Line 18-19:
There was: The amount of phenolic compounds contained in the alkaline hydrolysates accounted for 55.7 (soil, horizon Bsv) – 454% (roots) of their content in acid hydrolysates.
There is: The amount of phenolic compounds contained in the alkaline hydrolysates accounted for 55.7% (soil, horizon Bsv) – 454% (roots) of their content in acid hydrolysates.
Line 67-68:
There was: Assaying the contents of phenolic compounds in soils facilitates the evaluation of the degree of plant residue decomposition in soil [8,9,10,11,12,13].
There is: Assaying the contents of phenolic compounds in soils facilitates the evaluation of the degree of plant residue decomposition in soil [8-13].
Line 86:
There was: … in the extracts from alkaline re-hydrolysates the significantly higher contents ….
There is: … in the extracts from alkaline re-hydrolysates significantly higher contents …..
Line 131-134:
The contents of the other compounds, namely p-CO, SA, FEA, PA, CHA, p-HA and SYR, in the extracts from alkaline re-hydrolysates accounted for 96.4, 83.8, 79.1, 76.0, 75.4, 69.6 and 65.7% of the total of both types of hydrolyses, respectively.
The data referred to in the above sentence is not provided in the tables. The values in the tables are expressed in μg/g, and those provided in the sentence present a breakdown (in percentage) of the phenolic compounds extracted from alkaline re-hydrolysates in the pool of their content in both hydrolysates (acid hydrolysis and alkaline re-hydrolysis). The Authors refer to those values to stress the importance of alkaline re-hydrolysis in the extraction of p-CO, SA, FEA, PA, CHA, p-HA and SYR.
line 161-163:
There was: The contents of phenolic compounds recorded in the samples of plant and soil material have unambiguously demonstrated that, additionally, the application of alkaline hydrolysis after acid hydrolysis considerably increased the amount of the phenolic compounds extracted.
There is: The contents of phenolic compounds recorded in the samples of plant and soil material have demonstrated that the application of alkaline hydrolysis after acid hydrolysis considerably increased the amount of the phenolic compounds extracted.
Line 180:
The share of FEA, p-CO and SA in alkaline re-hydrolysates… the term “the share” has been correctly used.
Line 211:
There was: … the compounds allows for determining their origin [8,9,12,13,14,15].
There is: … the compounds allow for determining their origin [8,9,12-15].
Line 327-328:
There was: For statistical calculations, the Excel spreadsheet and Statistica MS 2012 package were applied.
There is: For statistical calculations, the Excel spreadsheet and Statistica MS 2012 package were used.
Reviewer 2 Report
The manuscript has been corrected with various substantive issues (that's good), but, in my opinion, it is still poorly read - I recommend further changes by a native speaker.
Citing authors 'As already reported earlier, the amounts of free phenols are inconsiderable and, as compared to soils, also inconsiderable and so this assay has been disregarded', I can accept the authors' explanation that the level of free phenolic compounds in soil samples is low (I do not have much experience with this material) however I can not accept it concerning plant material (some plants such as grass species contain high levels of free forms of phenolic acids such as chlorogenic acids).
Specific comments are included in the attached pdf file.

Author Response
Dear Reviewer,
The paper's Authors wish to thank for all the precious comments and guidelines.
I hope that the corrections introduced are satisfactory. All the changes have been marked in the text.
With best regards,
Magdalena Banach-Szott
Line 12-13:
There was: The study included the plant material samples (hay, sward and roots) and soils - Albic Brunic Arenosol (horizon A, AE and Bsv).
There is: The study included the plant material (hay, sward and roots) and soil - Albic Brunic Arenosol (horizon A, AE and Bsv) samples.
Line 34-37:
There was: hydroxybenzoic acid and its derivatives, e.g. dihydroxybenzoic, protocatechuic (PA), salicylic (SA), syringic (SYR), vanillic acids (VA);
– hydroxycinnamic acid and its derivatives, e.g. cinnamic, ferulic (FEA), p-coumaric (p-CO), caffeic acids (CA).
There is: hydroxybenzoic acid and its derivatives, e.g. dihydroxybenzoic, protocatechuic (PA), salicylic (SA), syringic (SYR), vanillic (VA) acids;
– hydroxycinnamic acid and its derivatives, e.g. cinnamic, ferulic (FEA), p-coumaric (p-CO), caffeic (CA) acids.
Line 38-40:
There was: Another group of phenolic compounds are depsides the molecule core of which includes an ester bond, e.g., chlorogenic acid (CHA) is produced as a result of the combination of the carboxyl group of caffeic acid with phenolic group of quinic acid [2].
There is: Another group of phenolic compounds are the molecule core of which includes an ester bond, e.g., chlorogenic acid (CHA) is produced as a result of the combination of the carboxyl group of caffeic acid with phenolic group of quinic acid [2].
Line 87-90:
There was: In the extracts from acid hydrolysates and alkaline re-hydrolysates from the plant material and soil samples, 11 phenolic aldehydes and phenolic acids have been identified (Table S1)
There is: In the extracts from acid hydrolysates and alkaline re-hydrolysates from the plant material and soil samples, 11 phenolic aldehydes and phenolic acids have been identified: protocatechuic acid (PA), p-hydroxybenzoic acid (p-HA), chlorogenic acid (CHA), vanillic acid (VA), syringic acid (SYR), caffeic acid (CA), vanillin (VAN), syringaldehyde (SYAL), p-coumaric acid (p-CO), ferulic acid (FEA) and salicylic acid (SA) (Table S1)
Line 202-207:
This paper determines the share of vanillyl, syringyl and cinnamyl compounds from the amount of phenolic compounds produced as a result of acid hydrolysis and the sum of hydrolyses the plant material and soil were exposed to; the content of phenolic compounds in lignins was not assayed. The methods are used to determine the total content of phenolic compounds in the plant and in soil.
As given in the Introduction to the paper, “Phenolic acids and phenolic aldehydes represent a group of polyphenols, common in plant tissues. They are secondary metabolites with a varied chemical composition and biological properties. In plants they are mostly found in bound forms, as esters or glycosides, as lignin and tannins hydrolysing components. As seen from the literature reports, vanillyl, syringyl and cinnamyl compounds and, mostly their ratios in lignins are plant-species-specific. For instance, as reported by Kovaleva and Kovalev, the lignins of herbaceous plants contain higher amounts of cinnamyl compounds than the contents of vanillyl and syringyl compounds. The lignins of grasses and meadow herbs can contain from 4- to 6-fold more cinnamyl compounds than the lignins of trees. The comparison provided in this paper shows only that the V:S:C ratio for the plant material is similar to the one characteristic for meadow plant lignins; it does not refer to their share only and exclusively in lignins.
The research was performed in the soil-plant system and so the plant material with specific values of the V:S:C ratio is the source of organic matter of soils and it undergoes the processes of decomposition (mineralization and humification). The consequence of the transformations of fresh organic matter in soil is a change in the content of phenolic compounds from the hydrolysis as well as the V:S:C ratio. And, similarly as for the plant material, also for the soil samples it has been demonstrated from the literature reports that the direction of changes in the V:S:C parameter is the same as in lignins.
Line 226-228:
There was: Interestingly, both for the plant material samples and for the soil samples, no significant differences in the content of vanillyl and syringyl compounds were found, whereas cinnamyl compounds were dominant.
There is: Interestingly, regardless of the sample type, cinnamyl compounds were dominant, and no significant differences in the content of vanillyl and syringyl compounds were found (Figure 2 and 3).
Figure 3 presents the results for the hydrolysis sum (acid and alkaline re-hydrolysis) and so it is impossible to draw a direct conclusion about the dominance of one of the hydrolyses.
Line 255-259:
There was: HCl, NaOH, ascorbic acid, EDTA, ethyl acetate: p.a., provided by Avantor Performance Materials, Poland.
Phenolic compounds, analytical standard: protocatechuic acid, p-hydroxybenzoic acid, chlorogenic acid, caffeic acid, vanillin, ferulic acid, salicylic acid provided by Sigma-Aldrich; vanillic acid, syringic acid, syringaldehyde, p-coumaric acid provided by Fluka
There is: HCl, NaOH, ascorbic acid, EDTA, ethyl acetate (all of analytical grade), and acetonitrile, methanol, and acetic acid (all of HPLC grade) were purchased from Avantor Performance Materials (Poland).
Analytical standards of phenolic compounds were purchased either from Sigma-Aldrich (Steinheim, Germany) - protocatechuic, p-hydroxybenzoic, chlorogenic, caffeic, ferulic and salicylic acids, and vanillin; or from Fluka (Steinheim, Germany) - vanillic, syringic and p-coumaric acids, and syringaldehyde.
Line 275-276:
There was: The samples were taken from the plot where irrigation had been stopped since 1996, referred to by the founders as the Green Meadow. The plant material (hay, sward and roots) and soil
There is: The samples were taken in 2011 from the plot where irrigation had been stopped since 1996, referred to by the founders as the Green Meadow. The plant material (hay, sward and roots) and soil
Line 291-292:
There was:
–acid hydrolysis with a HCl solution;
–alkaline re-hydrolysis performed after acid hydrolysis (Figure 5).
There is:
–acid hydrolysis with a 6 M HCl solution;
–alkaline re-hydrolysis with a 10 M NaOH solution.
The details of extraction procedures are given in Figure 5.
Figure 5 has been supplemented with the data suggested by the Reviewer.
Line 303:
The supplier of the Bionacom Velocity STR has been added.
Line 308-309
The Authors have applied three independent chromatographic runs for each sample.
Line 320-321
There was: The reference response curve for each phenolic compound was a linear regression with correlation coefficients > 0.98.
There is: The reference response curve for each phenolic compound was a linear regression with regression coefficients > 0.98. The detailed data is provided in Table 5.
Table 5. Regression coefficient, LOD and LOQ of phenolic compounds reference
|
Name of the standard |
Regression coefficient R2 |
LOD μg/mL |
LOD μg/mL |
|
Protocatechuic acid |
0.9831 |
0.22 |
0.67 |
|
p-Hydroxybenzoic acid |
0.9801 |
0.41 |
1.23 |
|
Chlorogenic acid |
0.9806 |
0.24 |
0.71 |
|
Vanillic acid |
0.9836 |
0.25 |
0.75 |
|
Syringic acid |
0.9863 |
0.35 |
1.04 |
|
Caffeic acid |
0.9817 |
0.26 |
0.78 |
|
Vanillin |
0.9865 |
0.17 |
0.51 |
|
Syringaldehyde |
0.9818 |
0.15 |
0.45 |
|
p-Coumaric acid |
0.9875 |
0.30 |
0.91 |
|
Ferulic acid |
0.9858 |
0.23 |
0.70 |
|
Salicylic acid |
0.9802 |
0.24 |
0.73 |
LOD Limit of detection, LOQ limit of quantification.